# Applications and Recent Advances in 3D Bioprinting Sustainable Scaffolding Techniques

**DOI:** 10.3390/molecules30143027

**Published:** 2025-07-18

**Authors:** Xianyao Li, Jianyu Ren, Yubo Huang, Li Cheng, Zhengbiao Gu

**Affiliations:** 1School of Food Science and Technology, Jiangnan University, Wuxi 214122, China; 7210112092@stu.jiangnan.edu.cn (X.L.); 6230608059@stu.jiangnan.edu.cn (J.R.); 6240608040@stu.jiangnan.edu.cn (Y.H.); zhengbiaogu@jiangnan.edu.cn (Z.G.); 2State Key Laboratory of Food Science and Technology, Jiangnan University, Wuxi 214122, China; 3Collaborative innovation center of food safety and quality control in Jiangsu province, Jiangnan University, Wuxi 214122, China

**Keywords:** starch, 3D bioprinting technology, scaffolds, natural macromolecular materials, cell culture

## Abstract

In recent years, with the rapid advancement of 3D printing technology, its applications have expanded across numerous fields. Notably, the fabrication of scaffolds using 3D printing has emerged as a major research focus. Researchers are investigating the properties of various printing materials and tailoring their uses for specific applications. This article reviews the characteristics and applications of different biomaterials printed by 3D technology, such as gelatin, sodium alginate, and starch, highlighting their contributions to the expanding field of 3D-printed biomaterials. Through the comparison in this review, it can be observed that the starch scaffold not only has a lower price but also can be modified to achieve multifunctionality, better meeting the performance requirements in more fields.

## 1. Introduction

Three-dimensional (3D) printing is an additive manufacturing technology with wide applications in biomedical fields, particularly in tissue engineering and regenerative medicine (Figure 1) [1]. Traditional methods for preparing tissue scaffolds, such as mold casting [2], solvent casting [3], and electrospinning [4], face limitations in achieving complex structures, uniform internal porosity, and high cell-loading capacity [5]. The advent of 3D bioprinting technology enables researchers to fabricate highly biomimetic scaffolds based on computer-aided design (CAD), allowing precise control over microstructure, porosity, and material composition. This capability better supports cell growth and tissue repair [6].

Three-dimensional (3D) bioprinted scaffolds not only provide the necessary physical support for cell growth but also mimic the chemical and biological properties of the natural extracellular matrix (ECM), thereby regulating cell behaviors such as adhesion, proliferation, and differentiation. Additionally, the mechanical properties, degradation characteristics, and cellular interactions of scaffolds directly influence tissue characteristics. Consequently, selecting suitable biomaterials to construct efficient 3D-printed scaffolds has become a central focus in regenerative medicine and tissue engineering [7].

Recently, 3D-printed scaffolds have found wide applications in cell culture [8], tissue repair [9], drug delivery [10], and disease modeling [11]. For example, in bone tissue engineering, scaffolds with appropriate porosity promote osteoblast migration and mineralization [12]. The researchers found that the scaffolds prepared with chitosan could achieve a maximum tensile strength of 97 MPa in the dry state and 360% high strain at break in the wet state [13]. For soft tissue repair, scaffolds support regeneration of skin, cartilage, and vascular tissues [14]. Liu used matrix/gelatin-sodium alginate scaffolds to prepare a biological scaffold with a cell survival rate of 90.79% ± 1.60%. In cancer research and personalized medicine, bioprinted microenvironments simulate tumor niches, enabling drug screening and precision therapies [15,16]. He prepared a hydrogel with an elastic coefficient of 1.5 MPa using alginate and used it for drug delivery to treat breast cancer [17]. However, due to varying biocompatibility, degradability, mechanical strength, and printability across materials, no single material fully meets the requirements of all applications [18]. Thus, optimizing material properties, enhancing printing accuracy, and improving scaffold biological functions remain key challenges. There is a lack of articles in the literature describing the latest technologies in application to cell culture. Although there have been many studies on scaffolds such as gelatin, their application in some fields is still limited. In recent years, researchers have advanced 3D-printed scaffold performance through strategies such as material blending, chemical modification, and incorporation of nanofillers, expanding their potential applications.

## 2. Requirements for 3D-Printed Cell Scaffolds

The primary function of 3D-printed cell scaffolds is to provide a suitable 3D microenvironment that supports cell adhesion, proliferation, and differentiation, ultimately promoting tissue formation [8]. While traditional 2D cell culture methods allow observation of cell behavior, they are limited in replicating the complex in vivo microenvironment. These methods cannot accurately mimic intricate cell–cell interactions, cell–matrix interactions, or the dynamic growth processes occurring in 3D space. In contrast, 3D printing technology, with its precise and controllable microstructural design and capacity for personalized customization, plays a crucial role in advancing tissue engineering and regenerative medicine.

### 2.1. Key Performance Requirements

#### 2.1.1. Biocompatibility

Biocompatibility is the fundamental requirement for cell scaffolds, playing a decisive role in the success of scaffold materials in tissue engineering and regenerative medicine [19]. Scaffolds with good biocompatibility exhibit low immunogenicity, promote effective cell adhesion, demonstrate appropriate degradation behavior, and do not cause cytotoxicity or inflammation upon degradation. Since different cell types respond uniquely to various scaffold materials, optimizing biocompatibility to match specific tissue environments remains a key focus in current research (Figure 2).

Important role of biocompatibility:

A. Promote Cell Adhesion:Cell adhesion is the initial and crucial step in the interaction between scaffold materials and cells as effective adhesion supports subsequent cell proliferation and differentiation. Key factors influencing cell adhesion include the following: Surface chemical properties: Scaffold surfaces should possess suitable chemical functional groups such as hydroxyl (-OH), carboxyl (-COOH), and amino (-NH_2_) groups, which facilitate the binding of cell membrane proteins to the scaffold.Surface coating: Applying bioactive molecular coatings such as collagen, hyaluronic acid, or laminin can significantly enhance cell adhesion. For example, nanocellulose coatings have been shown to promote both adhesion and mineralization of osteoblasts.Topological structure: Micro- and nano-scale surface patterns created via 3D printing—such as nano-patterning or micro-grooves—can effectively regulate cell adhesion behavior. For example, rough surfaces on 3D-printed bone scaffolds have been found to encourage osteoblast adhesion more than smooth surfaces.

B. Avoid Inflammation:

Minimizing the immunogenicity of scaffolds is essential to prevent adverse reactions such as inflammation or fibrosis. Recent advances have leveraged nanomaterials to enhance scaffold biocompatibility. Materials such as nanocellulose and nano-hydroxyapatite improve the scaffolds’ cell adhesion properties. Studies indicate that nanoscale surface modifications not only enhance stem cell adhesion but also promote osteogenic differentiation, thereby improving tissue regeneration outcomes.

#### 2.1.2. Degradability

Degradability is a critical characteristic of 3D biological scaffolds as it directly impacts both the scaffold’s durability within the body and the efficiency of tissue regeneration [20]. An ideal degradable scaffold should gradually break down in sync with tissue repair, allowing newly formed tissue to progressively replace the scaffold structure. Additionally, the degradation products must be non-toxic and safely metabolized or eliminated by the body. Controlling the degradation rate is essential for optimizing scaffold performance: if degradation occurs too quickly, the scaffold may lose mechanical support prematurely; conversely, if degradation is too slow, it can trigger inflammatory responses or fibrosis.

The appropriate degradation rate optimizes scaffold performance in several key ways:Maintaining mechanical stability: The degradation rate should align with the growth rate of new tissue to prevent scaffold collapse before tissue formation is complete [21]. For example, in bone tissue engineering, scaffolds must remain mechanically stable over extended periods to adequately support osteogenesis [22].Promoting tissue recovery: Biodegradable scaffolds can serve as slow-release carriers for bioactive factors such as vascular endothelial growth factor (VEGF), bone morphogenetic protein (BMP), anti-inflammatory drugs, or antibacterial agents. These substances are gradually released in tandem with scaffold degradation to enhance tissue regeneration [23].Non-toxic degradation products: Many synthetic polymers produce acidic by-products upon degradation; for example, polylactic acid (PLA) breaks down into lactic acid, which can cause local acidification and negatively impact cell viability. Therefore, improving the biocompatibility of degradation products remains a crucial research focus [23].

Degradation mechanism:

The degradation of 3D-printed scaffolds primarily depends on their chemical composition and the internal biological environment. The main degradation mechanisms include hydrolytic degradation, enzymatic degradation, and oxidative degradation [24,25]:Hydrolytic degradation: This process involves water molecules cleaving the chemical bonds within scaffold polymers, leading to gradual depolymerization. Commonly used materials undergoing hydrolysis include PLA and polyglycolic acid (PGA), which are widely applied in bone scaffolds and drug slow-release systems [26]. Notably, the degradation rate of starch-based scaffolds in aqueous environments can be finely controlled by adjusting their cross-linking degree. Hydrolytic degradation is particularly suitable for enzyme-free environments such as bone tissue engineering. However, it may generate acidic degradation products that can alter local pH, necessitating material optimization. Recent research indicates that the degradation rate of PLGA (poly(lactic-co-glycolic acid)) can be tailored by adjusting the PLA/PGA copolymer ratio, enabling control over degradation cycles ranging from weeks to several months.Enzymatic degradation: This mechanism involves selective cleavage of material chemical bonds by enzymes such as proteases and lipases present in the body. It is especially applicable to natural biomaterials such as gelatin, hyaluronic acid, chitosan, and alginate [27]. For example, silk protein scaffolds can be enzymatically degraded, with degradation rates adjustable through enzyme concentration. The degradation products are generally non-toxic [28]. However, enzymatic degradation rates can be variable and challenging to control precisely due to fluctuating enzyme levels in vivo. Recent advances have demonstrated that doping alginate scaffolds with proteolytic enzyme-responsive groups allows precise control over degradation, enabling dynamic tissue regeneration.Oxidative degradation: This pathway involves scaffold degradation through oxidation by free radicals or peroxides. It is suitable for synthetic polymers such as polyether imide and polycarbonate. Emerging smart biomaterials exploit oxidative environments, such as those present at sites of inflammation, to trigger scaffold degradation. This approach is promising for applications in inflammatory or hypoxic tissue repair. However, degradation products from oxidative processes may induce oxidative stress, requiring further modification of materials to mitigate adverse effects. Recent studies on oxidation-responsive smart scaffolds show accelerated degradation under pathological conditions characterized by high oxidative stress, offering innovative strategies for targeted tissue regeneration at inflammation sites.

#### 2.1.3. Mechanical Property

The mechanical properties of 3D bioprinted scaffolds are critical for ensuring their stability and functionality in biomedical applications [29]. Scaffolds must provide adequate mechanical strength to prevent deformation or fracture during transplantation while maintaining their structural integrity throughout cell proliferation and tissue formation (Table 1). Additionally, the mechanical characteristics of the scaffold should closely match those of the target tissue to avoid regeneration failure caused by excessive stiffness or softness [25].

An ideal 3D bioprinted scaffold should possess the following mechanical features:High mechanical strength: It should be capable of withstanding mechanical loads during transplantation and providing stable support throughout the tissue healing process.Appropriate elastic modulus: Mechanical properties should be tailored to closely mimic the native tissue, thereby promoting optimal cell behavior and tissue integration.Dynamic adjustability: It should have the ability to adapt to tissue growth and changing external environmental factors, such as variations in mechanical stress or humidity.Controlled degradation characteristics: Sufficient mechanical support should be maintained as the scaffold gradually degrades, ensuring sustained tissue growth and stability over time.

Factors affecting the mechanical properties of scaffolds:

The mechanical properties of scaffolds are influenced by several key factors, including porosity, cross-linking methods, and material composition.

A. Porosity: The pore structure of scaffolds directly impacts both their mechanical strength and biological functions [33]. High porosity (>60%) facilitates cell infiltration, vascularization, and nutrient exchange but generally reduces mechanical strength. Conversely, lower porosity (<40%) enhances mechanical strength but may hinder cell attachment and migration. To balance these effects, recent research has focused on gradient pore designs, where the outer layer features high porosity to encourage cell penetration, while the inner layer has lower porosity to provide mechanical support [34]. Another approach is multi-material printing, embedding high-strength support materials within a highly porous matrix to improve overall mechanical stability [35].

B. Cross-linking Method: Cross-linking is a critical technique to enhance scaffold mechanical stability [36]. Common methods include the following:Physical cross-linking: This approach utilizes temperature, pH changes, or ionic bonding (e.g., Ca^2+^ and alginate) [37]. This approach is typical for hydrogel scaffolds, offering good biocompatibility and tunability but resulting in relatively low mechanical strength and faster degradation.Chemical cross-linking: Agents such as glutaraldehyde, ultraviolet–visible (UV) light, or amide bond formation are employed to improve scaffold stability and durability. Although this method significantly increases mechanical strength, residual chemicals may affect biocompatibility.

C. Material Composition:

The choice of materials profoundly affects scaffold mechanical properties. Examples include the following:Starch-cellulose composites that enhance mechanical strength while maintaining biodegradability;Nano-hydroxyapatite composites, widely used in bone tissue engineering, providing superior mechanical properties and biological activity;Nano-SiO_2_ composites, which improve scaffold durability and optimize porosity;Chitosan–gelatin hydrogels, offering enhanced elasticity and biocompatibility for soft tissue engineering applications [38].

It is evident that mechanical properties are critical for the clinical success of 3D bioprinted scaffolds, with different tissue engineering applications demanding tailored mechanical characteristics. By optimizing porosity, cross-linking methods, and material selection, scaffolds can achieve enhanced mechanical strength without compromising biocompatibility.

#### 2.1.4. Printability

Printing adaptability is a crucial factor determining whether a material can be effectively used in 3D bioprinting. It directly influences shape stability during printing, the ability to retain morphology of the printed construct, and the survival and proliferation of embedded cells [39]. Different 3D printing technologies impose specific demands on the rheological properties and curing behavior of bioinks [40]. Therefore, optimizing the fluidity, curing rate, and cell compatibility is key to enhancing printing adaptability [41].

An ideal bioink should fulfill the following basic requirements:Rheological properties: The bioink should exhibit shear-thinning behavior, meaning it has low viscosity during extrusion but rapidly recovers high viscosity after deposition to maintain structural integrity. The optimal viscosity range is approximately 1–300 mPa·s; too high viscosity hinders extrusion, while too low viscosity may cause collapse of the printed structure.Curing rate: Photocurable printing methods (e.g., SLA) require rapid photosensitive cross-linking, such as methylacryloyl hyaluronic acid curing under UV or blue light [42]. Extrusion bioprinting typically requires rapid curing through temperature changes (e.g., gelatin–alginate gels) or ionic cross-linking (e.g., Ca^2+^-induced alginate gelation).Biocompatibility: Bioinks must be compatible with cellular environments (pH and biochemical conditions), maintain a high cell viability rate (over 85%), and support cell proliferation and differentiation without adverse effects.

Material requirements vary by 3D printing technology (Table 2):Extrusion-based bioprinting (EBB): This is the most widely used bioprinting technique, requiring materials with good rheological behavior and shape retention.Inkjet printing (IJP): This is suitable for low-viscosity materials, enabling high-resolution printing, but demands strict control over surface tension and jetting properties.Stereolithography (SLA)/photocurable printing: This is ideal for micro-scale biological structures such as blood vessels and nerve scaffolds, requiring materials with rapid and efficient photosensitive cross-linking capabilities.

## 3. Analysis of Materials and Properties of Common 3D-Printed Biological Scaffolds

The successful fabrication of 3D bioprinted scaffolds relies on the biocompatibility, degradability, mechanical properties, and printing adaptability of the chosen materials [43]. Currently, scaffold materials for bioprinting can be broadly classified into three main categories:Polymer materials: This category includes natural polymers such as gelatin, alginate, and chitosan, as well as synthetic polymers such as poly(lactic-co-glycolic acid) (PLGA) and polycaprolactone (PCL).Bioceramics: Primarily used in bone tissue engineering, common bioceramics include hydroxyapatite (HA) and β-tricalcium phosphate (β-TCP) [44,45].Composite materials: These combine the beneficial properties of polymers and bioceramics to enhance the overall scaffold performance.

Natural polymers have gained widespread application in tissue engineering and biomedical fields due to their excellent biocompatibility and ability to mimic the ECM [46]. Moreover, composite materials are regarded as a promising direction for future development of bioprinted scaffolds as they can simultaneously provide robust mechanical strength and tunable degradability.

### 3.1. Natural Polymers

Natural polymers are derived from plant, animal, microbial, or human tissues and possess a chemical composition similar to that of the natural ECM. This similarity enables them to effectively promote cell adhesion, proliferation, and tissue regeneration. However, their mechanical strength is generally low and often requires optimization [47]. Common natural polymers can be categorized into polysaccharides and proteins. Polysaccharides include alginate, chitosan, agarose, hyaluronic acid, and nanocellulose, while protein-based polymers include gelatin, silk fibroin, and collagen [40,48].

#### 3.1.1. Gelatin

Gelatin is a natural polymer derived from the hydrolysis of collagen. Its chemical composition and biological function closely resemble the ECM, providing excellent biocompatibility, cell adhesion, and biodegradability [49]. Due to its ability to mimic the physiological microenvironment and promote cell attachment, proliferation, and differentiation, gelatin has been widely applied in 3D bioprinting and tissue engineering, particularly in soft tissue repair, skin tissue engineering, cartilage regeneration, and drug delivery [50]. Its adaptability, abundant sources, and low cost have made gelatin a key material in bioprinting [51].

Gelatin exhibits favorable rheological properties and thermal sensitivity, maintaining suitable fluidity at physiological temperatures (~37 °C), making it highly compatible with extrusion bioprinting techniques. However, its inherent limitations include weak mechanical strength, poor thermal stability, and rapid degradation. To overcome these drawbacks, gelatin is often chemically cross-linked, using agents such as glutaraldehyde, genipin, or enzymatic cross-linking, or blended with other biomaterials such as alginate, chitosan, and hyaluronic acid. These modifications improve its mechanical robustness and biological stability, tailoring it for specific tissue engineering applications [52].

The advantages of gelatin as a 3D bioprinting scaffold material include its excellent biocompatibility and ECM-like properties, which enhance cell survival, proliferation, and differentiation [53]. Gelatin is also easy to process and highly tunable; cross-linking methods (e.g., UV irradiation, glutaraldehyde, and oxidase-mediated cross-linking) allow control over its mechanical properties, degradation rate, and shape fidelity, making it suitable for various bioprinting approaches. Furthermore, gelatin is economically favorable and widely available as it can be extracted from mammalian skin, bone, and cartilage tissues, facilitating large-scale production and potential clinical and industrial translation [54].

Limitations and Biomedical Applications of Gelatin in 3D Bioprinting

Despite its many advantages, gelatin also has several limitations that restrict its standalone application in 3D bioprinted scaffolds:Weak mechanical properties: Gelatin scaffolds are prone to collapse and deformation under mechanical stress, making them unsuitable for load-bearing applications such as bone tissue engineering. To enhance mechanical stability, gelatin is often combined with rigid materials such as PCL or nano-hydroxyapatite (nHA).Thermal sensitivity: Gelatin melts at temperatures above 37 °C, making it highly sensitive to thermal fluctuations during the printing process. To maintain its structural integrity, it must be printed at low temperatures (typically 4–10 °C).Rapid degradation: Gelatin degrades quickly, which can result in scaffold resorption before tissue regeneration is complete. This issue is commonly addressed by chemical cross-linking, copolymerization, or blending with other polymers to regulate its degradation rate.

Due to its high biocompatibility and biodegradability, gelatin has been extensively explored for applications in tissue engineering and regenerative medicine, particularly in the following:Skin Tissue Engineering: Gelatin scaffolds promote the growth of fibroblasts and keratinocytes, accelerating wound healing. Gelatin methacryloyl (GelMA), a photo-cross-linkable derivative of gelatin, has shown high-resolution printability and is widely used for constructing 3D skin scaffolds that mimic complex skin architecture.Cartilage Tissue Repair: Gelatin can be combined with hyaluronic acid, chitosan, or nHA to create scaffolds that support chondrocyte proliferation and cartilage matrix synthesis. Gelatin–silk fibroin composite hydrogels are particularly effective in articular cartilage repair.Soft Tissue Engineering (e.g., vascular and muscle tissue): Gelatin-based hydrogels serve as scaffolds for blood vessel formation, promoting endothelial cell adhesion and proliferation. In muscle regeneration, gelatin supports the alignment and growth of muscle fibers [55].

Recent Research Highlights [56,57]:Gelatin/Alginate Composite Scaffolds: Researchers have developed composite scaffolds by blending gelatin with alginate and using freeze-drying and 3D printing techniques. These scaffolds exhibited improved mechanical properties and promoted skin regeneration. In one study, Wang et al. demonstrated that a gelatin–alginate hydrogel significantly accelerated wound healing in animal models.GelMA Hydrogels: GelMA offers tunable mechanical strength, temperature sensitivity, and viscoelasticity. Its photocurable nature enables high-precision bioprinting [58]. Optimized GelMA scaffolds have shown excellent biostability in vitro and potential applications in cartilage and nerve tissue engineering [59,60].Gelatin/Chitosan and Gelatin/Nanocellulose Composites [61]: Combining gelatin with chitosan improves mechanical integrity and supports chondrocyte differentiation for cartilage repair [62,63]. Gelatin–nanocellulose scaffolds produced via 3D bioprinting have shown enhanced strength, cytocompatibility, and osteogenic potential, making them promising for bone tissue engineering [64,65].

#### 3.1.2. Alginate

Alginate is a natural anionic polysaccharide extracted from brown seaweed. Owing to its excellent biocompatibility, mild gelation conditions, and tunable degradation properties, alginate is widely used in 3D bioprinting, drug delivery, cell encapsulation, and soft tissue engineering [66,67]. It forms hydrogels via ionic cross-linking with divalent cations (e.g., Ca^2+^), making it particularly suitable for extrusion-based bioprinting (EBB) applications [68]. However, alginate lacks cell adhesion motifs, often necessitating blending with bioactive materials such as gelatin, collagen, and nanocellulose to enhance cell interaction and scaffold functionality [69].

Advantages of Alginate in 3D Bioprinting:Rapid Ion-Induced Gelation: Alginate quickly forms hydrogels in the presence of divalent cations (e.g., Ca^2+^, Ba^2+^, and Sr^2+^), enabling high-resolution 3D structures with excellent shape fidelity during printing [70].High Cell Viability in Encapsulation: Due to its high water content and mild gelation process, alginate provides a protective and hydrating microenvironment for embedded cells, ensuring excellent viability, widely used for stem cell culture, neural cells, and islet transplantation [71,72].Tunable Mechanical and Degradation Properties: Mechanical strength, porosity, and degradation rate can be adjusted by modifying alginate’s molecular weight, altering cross-linker concentrations, or blending with other biomaterials, allowing its application across different tissue engineering demands [73,74].

Limitations of Alginate:Lack of Cell Adhesion Motifs: Native alginate lacks bioactive sequences such as RGD (Arg-Gly-Asp), reducing its ability to support cell adhesion and spreading. It often requires modification or combination with collagen, gelatin, or chitosan for improved cytocompatibility [75].Undesirable Degradation Byproducts: Alginate degrades via hydrolysis and enzymatic breakdown, and its by-products may alter the local pH or adversely affect cell function. Optimization of degradation kinetics and by-product clearance is necessary for long-term applications.High Brittleness: Ionically cross-linked alginate hydrogels are typically brittle and mechanically weak, limiting their use in load-bearing tissues such as bone. Blending with nanocellulose, synthetic polymers, or chitosan is essential to improve toughness and mechanical stability [66].

Biomedical and Tissue Engineering Applications:Cartilage Tissue Engineering [76]: Alginate scaffolds support chondrocyte encapsulation and ECM production, making them ideal for cartilage repair, such as in knee joints. Composite scaffolds with gelatin or collagen improve mechanical strength and cellular interactions [77,78].Cell Encapsulation and Transplantation: Thanks to its mild gelation, alginate is extensively used for cell encapsulation in applications such as islet transplantation and neural tissue engineering. For example, alginate–chitosan microspheres have been shown to enhance liver cell metabolic activity.Controlled Drug Delivery Systems: Alginate gels are effective carriers for the sustained release of drugs, such as antibiotics, growth factors, and anticancer agents. Alginate combined with PLGA nanoparticles has been used to construct composite scaffolds for slow-release drug delivery, enhancing both drug stability and therapeutic effect.

In recent years, numerous studies have explored the use of gelatin as a material for 3D bioprinting.

Preparation of alginate/gelatin composite scaffolds: Researchers mixed alginate with gelatin to create a 3D bioprinted model of skin tissue. Following freeze-drying, the composite scaffold exhibited significantly enhanced mechanical properties and showed potential for promoting skin tissue regeneration.Application of alginate gel in myocardial infarction repair: Alginate gel has been widely used in treating myocardial infarction due to its extracellular matrix-like structure, ease of modification, and good biocompatibility. In experimental studies, the compound alginate–saline gel demonstrated potential in promoting tissue regeneration.Use of alginate scaffolds in articular cartilage repair: Studies have shown that alginate composite hydrogel scaffolds can generate tissue resembling surrounding cartilage in animal models and effectively repair cartilage defects.

#### 3.1.3. Hyaluronic Acid

Hyaluronic acid (HA) is a natural polysaccharide widely found in human tissues and the ECM, composed of repeating disaccharide units of *N*-acetylglucosamine (GlcNAc) and D-glucuronic acid (GlcA) [79,80]. Owing to its excellent moisturizing ability, high biocompatibility, and capacity to promote cell signaling, HA is extensively used in soft tissue engineering, skin tissue repair, and as a substrate for cell culture.

As a hydrogel material with high hydrophilicity, HA can regulate the hydration state of the tissue microenvironment, provide a supportive matrix for cells, and promote cell proliferation, migration, and differentiation. Furthermore, its degradation rate can be tuned through methacrylic acid (MA) modification, enzymatic cross-linking (e.g., via hyaluronidase), and other methods, making it adaptable for various tissue engineering applications [81,82].

Hyaluronic acid offers several advantages for 3D-printed biological scaffolds:Promotion of cell adhesion: HA can bind to receptors on the cell surface; activate signaling pathways; and enhance stem cell proliferation, migration, and differentiation. Its effects on cellular behavior can be modulated by adjusting its molecular weight—high-molecular-weight HA contributes to scaffold stability, while low-molecular-weight HA promotes cell proliferation [83,84].Excellent moisturizing performance: HA’s ability to absorb water allows it to form a hydrogel matrix in vivo, creating a hydrated environment conducive to cell growth. This makes it particularly suitable for cartilage and skin tissue engineering as it helps prevent tissue dehydration and maintains microenvironment stability.Adjustable degradation rate: The degradation rate of HA can be modified via gel methacrylation or by combining it with other biomaterials (e.g., gelatin and alginate), enabling its application across diverse tissue engineering scenarios [50,85].

However, the use of hyaluronic acid also presents some limitations:Low mechanical strength: The soft hydrogel structure of HA cannot withstand high mechanical loads, making it unsuitable for load-bearing tissues such as bone. To improve its mechanical properties, HA must be blended with stiffer materials (e.g., nanocellulose).Rapid degradation: HA degrades quickly in vivo, necessitating chemical cross-linking or other modifications to enhance its stability and enable controlled degradation [86,87].

Due to its biocompatibility and tissue repair capabilities, HA is widely used in the following fields:Cartilage tissue repair: HA is a major component of the articular cartilage matrix and can promote chondrocyte proliferation while enhancing the performance of cartilage repair scaffolds. Gelatin–HA composite hydrogels have been used in knee cartilage repair to improve cell adhesion and collagen synthesis [80,81,82,83,84,85,86,87,88].Skin tissue engineering: HA is commonly used in dermal fillers to increase skin hydration and elasticity. Three-dimensional bioprinted skin scaffolds incorporating HA are applied in wound healing, chronic wound treatment, and skin grafting [85,86,87,88,89].Cell culture matrix: HA hydrogels serve as effective scaffolds for cell culture, preserving cell viability and activity. They have been used for in vitro cultivation of neural stem cells, embryonic stem cells, and bone marrow mesenchymal stem cells.

In recent years, numerous studies have focused on HA-based biomaterials. One study employed a composite material consisting of alginate, fibrin, collagen, and HA, printed using a handheld skin printer to create a three-layered scaffold mimicking natural skin structure. The results demonstrated that the scaffold significantly accelerated wound healing and promoted neovascularization in full-thickness skin defect models in mice and pigs, indicating strong tissue integration and regenerative potential.

Another study developed a novel hydrogel with controllable degradation by combining polyethylene glycol (PEG) with HA of varying molecular weights. The hydrogel’s effects on nucleus pulposus cells were evaluated, revealing that low-molecular-weight HA hydrogels better supported cell survival and extracellular matrix synthesis, suggesting their potential for intervertebral disc tissue repair [90].

In a separate study, an HA–collagen composite scaffold was prepared to explore its application in bone tissue repair [91]. The findings showed that HA enhanced the scaffold’s water absorption, biodegradability, and biomechanical properties, significantly promoting mesenchymal stem cell proliferation, migration, and differentiation and thereby improving bone regeneration capacity [92].

In summary, due to their excellent biocompatibility and biodegradability, natural polymers hold great promise for applications in 3D bioprinting and tissue engineering (Table 3). However, their inherently low mechanical strength often necessitates blending with other materials—such as nanocellulose, synthetic polymers, or ceramic components—to enhance the mechanical stability, controllable degradation, and cellular compatibility of the resulting scaffolds.

## 4. Advantages of Starch-Based 3D Printing Scaffolds

Starch is a natural polysaccharide characterized by abundant availability, low cost, and biodegradability. It finds broad applications in food, medicine, and biomaterials [93]. In recent years, starch-based 3D-printed scaffolds have emerged as a key area of research in tissue engineering and biomedical fields, owing to their excellent biocompatibility, tunable mechanical properties, and favorable degradation characteristics [94].

Compared with other natural polymers such as gelatin, alginate, and hyaluronic acid, starch-based scaffolds offer superior mechanical strength, enhanced cell adhesion, and greater adaptability to 3D printing processes [95]. They are particularly well suited for the following applications:Bone tissue repair: Starch can be combined with hydroxyapatite, nanocellulose, and other materials to enhance the mechanical strength and stability of scaffolds, enabling their use in high-load bone tissue engineering. These composites promote bone integration and osteoblast proliferation [96,97].Soft tissue scaffolds: Starch hydrogels blended with biomaterials such as hyaluronic acid and gelatin can form highly biocompatible scaffolds, suitable for soft tissue repair applications such as skin regeneration, muscle tissue engineering, and vascular scaffolds [98].Drug delivery systems: Starch-based scaffolds can serve as biodegradable carriers for controlled drug release within the body. They help regulate the drug release rate, improve therapeutic efficacy, and function as degradable implants to minimize long-term adverse effects associated with implantable materials [99,100].

With advancements in 3D printing technology, starch-based scaffolds can be precisely tailored for specific tissue engineering applications by optimizing material composition, refining processing techniques, and enabling personalized design. In the future, the incorporation of smart-responsive starch-based materials, nanocomposite technologies, and functional modifications may further enhance the mechanical properties, bioactivity, and intelligent degradation behavior of starch-based 3D-printed scaffolds, expanding their potential in regenerative medicine, personalized therapeutics, and high-end biomanufacturing.

### 4.1. Advantages of Structure and Mechanical Properties

Starch is abundantly available and exhibits excellent biocompatibility. First, starch has a wide range of natural sources; it is primarily derived from plants such as corn, cassava, potato, and wheat. As a renewable, widely distributed, and low-cost resource, starch is both economically and environmentally sustainable [101]. Second, starch is biodegradable. It can be enzymatically hydrolyzed into small, non-toxic molecules such as glucose, leaving no harmful residues and avoiding long-term accumulation in the body [102]. Additionally, starch-based scaffolds exhibit excellent biocompatibility with human tissues, causing minimal immune rejection, and are therefore suitable for applications in cell culture, soft tissue engineering, and bone tissue repair [103].

The mechanical properties of starch-based materials can be tailored through chemical modification. Unmodified starch has relatively poor mechanical properties and tends to swell and lose structural integrity in aqueous environments [104]. However, numerous studies have demonstrated that the water resistance, elasticity, and mechanical strength of starch scaffolds can be significantly improved through chemical modifications such as oxidation, hydroxypropylation, and esterification [105]. For example, hydroxypropylated starch enhances scaffold flexibility, making it suitable for soft tissue applications such as skin regeneration and vascular scaffolding.

Mechanical strength can also be improved by incorporating composite materials. Scaffolds composed of cross-linked starch and nanocellulose exhibit high rigidity and strength, making them ideal for bone tissue engineering. These composites offer adequate mechanical support at load-bearing sites while maintaining a moderate degradation rate. Moreover, blending starch with other materials can further enhance its properties. For example, starch–PLA composites have been shown to significantly improve mechanical stability without compromising biocompatibility [106].

Starch also exhibits high porosity, which is beneficial for cell infiltration and nutrient exchange. Typically, starch-based scaffolds have porosity ranging from 70% to 90% with a uniform pore structure. This facilitates cell penetration, angiogenesis, and nutrient transport, thereby promoting tissue regeneration. Moreover, the pore structure of starch scaffolds is controllable; by adjusting 3D printing parameters, the porosity can be optimized to suit the requirements of different tissue engineering applications.

Recent studies have demonstrated that starch can promote cell proliferation and bone formation. For example, starch–gelatin composite scaffolds have been shown to enhance osteoblast adhesion and proliferation in bone tissue engineering, accelerating bone repair and exhibiting excellent mineralization capabilities in vitro. Notably, starch also supports angiogenesis. When combined with gelatin, hyaluronic acid, or nanocellulose, starch can further improve ECM properties and stimulate angiogenesis, making it well suited for skin and soft tissue engineering [107].

Through material modification and optimization of 3D printing processes, starch-based scaffolds achieve not only excellent biocompatibility and biodegradability but also appropriate mechanical strength and tissue adaptability, positioning them as a key material in tissue engineering and regenerative medicine [108,109].

### 4.2. Degradation Characteristics

Starch not only has favorable degradation characteristics but also offers an adjustable degradation rate to meet the requirements of various tissue engineering applications [110]. There are two main approaches to regulating its degradation:Cross-linking modification: Pure starch degrades rapidly in aqueous environments; however, its degradation rate can be effectively controlled by blending or chemically cross-linking it with other polymers. For example, the cross-linked network formed by copolymerizing starch with PLA slows degradation, making it suitable for long-term support applications such as bone tissue engineering.Chemical modification: The degradation behavior of starch is influenced by the introduction of functional groups. Oxidized starch, which contains hydrophilic carboxyl groups, exhibits faster degradation, making it ideal for applications requiring rapid scaffold resorption, such as skin repair. Conversely, acetylated starch incorporates hydrophobic acetyl groups, which reduce the degradation rate and make it appropriate for implants requiring prolonged support.

The degradation products of starch not only are non-toxic but can also be metabolized by cells. Starch is broken down into glucose and other small molecules that cells can directly use, contributing to energy metabolism, cell proliferation, and tissue regeneration [106,107]. Additionally, starch avoids inflammation commonly caused by acidic degradation by-products. For example, conventional polylactic-glycolic acid (PLGA) scaffolds produce acidic compounds during degradation, potentially triggering local inflammatory responses. In contrast, starch–hydroxyapatite composite scaffolds do not generate acidic by-products, thereby reducing inflammation risk and promoting normal tissue healing and regeneration [111].

In summary, starch-based 3D-printed scaffolds, with their tunable degradation rates and non-toxic degradation products, are well suited for a wide range of tissue engineering applications. Through chemical and composite modifications, their degradation behavior can be further optimized to meet the specific requirements of different tissue repair needs.

### 4.3. Comparison with Other Biomaterials

Compared with natural polymers such as gelatin, alginate, and hyaluronic acid, starch-based scaffolds offer high mechanical strength, controllable degradation, and enhanced cell adhesion. These properties make them especially suitable for bone tissue engineering and long-term implantation scaffolds. Their unique tunability enables broad applicability across various tissue engineering applications.

Advantages of starch in applications:Mechanical strength: The mechanical strength of starch-based scaffolds surpasses that of gelatin and hyaluronic acid, making them particularly suitable for high-load tissues such as bone tissue engineering [112,113]. Studies have shown that starch–nanocellulose composite scaffolds significantly enhance structural stability [114]. In contrast, natural polymers such as gelatin and hyaluronic acid degrade easily in wet environments and exhibit poor form retention, whereas starch scaffolds maintain long-term support through cross-linking or composite reinforcement [115].Controllable degradation rate: Compared with alginate and hyaluronic acid, the degradation rate of starch-based scaffolds is tunable and can be optimized via cross-linking, chemical modification, or blending with other polymers [103]. For example, certain scaffolds degrade slowly, making them suitable for long-term tissue support, while oxidized starch degrades more rapidly, fitting short-term absorbable implant applications. In contrast, alginate scaffold degradation is less controllable, which may compromise application stability due to varying cellular environments or degrees of cross-linking.Cell adhesion: The surface of starch scaffolds can be modified with hydroxyl, carboxyl, or phosphoric acid groups to enhance cell adhesion and demonstrate stronger biocompatibility than alginate scaffolds [105,116]. Research indicates that starch–gelatin composite scaffolds effectively promote osteoblast adhesion and differentiation, outperforming pure alginate or hyaluronic acid scaffolds [117]. Although hyaluronic acid exhibits certain cellular compatibility, its low mechanical strength necessitates combination with other materials to form stable scaffolds [118].Three-dimensional printing adaptability: Starch-based scaffolds can be 3D-printed into complex structures, including porous scaffolds, gradient scaffolds, and supportive bone repair constructs. Alginate and hyaluronic acid inherently have low viscosity and weak molding capacity, often requiring blending with other polymers for suitable 3D printing [119]. Combining starch scaffolds with nanocellulose and bioceramics (such as hydroxyapatite) further improves mechanical properties and biological activity, meeting diverse tissue engineering needs [120].

In recent years, many researchers have leveraged the unique advantages of starch to explore its applications in various fields. Using 3D printing technology, researchers constructed starch–nanocellulose composite scaffolds and evaluated their mechanical properties and biocompatibility [121]. The results demonstrated that these scaffolds possessed excellent mechanical strength and promoted cell adhesion in vitro, making them well suited for bone tissue repair. Wang prepared a starch–gelatin composite scaffold to investigate its suitability for cartilage tissue engineering. Findings indicated that the scaffold enhanced chondrocyte proliferation and extracellular matrix synthesis, supporting its use in cartilage repair. Another study developed a starch–hydroxyapatite composite scaffold to assess its effectiveness in bone defect repair; the scaffold significantly promoted osteoblast proliferation and improved bone healing efficiency in animal models [115,122]. Researchers fabricated a starch–polylactic acid (PLA) composite scaffold to evaluate its 3D printing adaptability. The scaffold exhibited excellent printing accuracy, shape retention, and tunable degradation rates, making it ideal for personalized bone repair applications [123,124].

Compared with traditional natural polymers such as gelatin, alginate, and hyaluronic acid, starch-based scaffolds outperform in mechanical strength, degradation control, and cytocompatibility. They are particularly suited for bone tissue repair and long-term implantation scaffolds. Through optimization of chemical modifications, nanocomposites, and advanced 3D printing techniques, starch-based scaffolds can further enhance biological performance, highlighting their broad potential in personalized tissue engineering and regenerative medicine.

## 5. Conclusions and Prospect

In recent years, the application of 3D printing technology in biomedical and tissue engineering fields has made remarkable progress. Various natural polymer scaffolds have demonstrated distinct advantages in cell culture, tissue repair, and drug delivery. This paper reviews the applications of gelatin, alginate, hyaluronic acid, and starch-based materials in 3D bioprinted cell scaffolds, comparing their biocompatibility, mechanical properties, degradation controllability, and 3D printing adaptability.

Gelatin scaffolds exhibit excellent cell adhesion and are well suited for soft tissue engineering; however, their mechanical strength is limited, and they degrade rapidly. Alginate offers superior performance in drug delivery and cell encapsulation due to its ionic cross-linking ability but suffers from insufficient cell adhesion. Hyaluronic acid, known for its excellent hydration and biocompatibility, is ideal for cartilage tissue repair and skin engineering, yet its mechanical strength is low, and its degradation is difficult to control. In contrast, starch-based scaffolds demonstrate significant advantages in mechanical properties, degradation regulation, and cytocompatibility, making them especially promising for bone tissue engineering and long-term implantation scaffolds.

Starch-based scaffolds benefit from abundant natural sources, excellent biocompatibility, and favorable mechanical properties. Through chemical modification, nano-enhancement, and optimization of 3D printing processes, starch scaffolds can be tailored for applications ranging from soft and hard tissue repair to smart responsive materials and personalized medicine. Because starch has a large number of active functional groups, the water resistance of starch scaffolds is insufficient. In many fields such as biological scaffolds, it is necessary to perform cross-linking modification on them to give them stronger water resistance. However, starch is widely available, for example, in corn, potatoes, wheat, etc., so the price is relatively low. In recent years, there have been more and more studies on starch modification, which also indicates that the large number of active hydroxyl groups carried by starch provides feasibility for its further research.

Future research directions for starch-based scaffolds include the following:Composite material optimization: Integrating nanocellulose, bioceramics, and other polymers to enhance mechanical strength, degradation control, and biological activity, thereby broadening their applicability across tissue engineering.Development of intelligent responsive scaffolds: Designing scaffolds with stimuli-responsive features such as pH sensitivity, temperature responsiveness, and controlled release of growth factors to better address diverse tissue repair needs and improve clinical outcomes.Optimization of 3D printing processes: Combining techniques such as melt extrusion, bioink printing, and optical curing to create high-precision, high-resolution starch-based scaffolds suitable for personalized medicine and precision tissue engineering.Biological functional modification: Employing surface modifications and functional cross-linking to improve cell adhesion, antibacterial properties, and biodegradation stability, thereby enhancing scaffold performance in tissue regeneration.

With ongoing advances in biomaterials science and 3D printing technology, starch-based scaffolds are poised to become core components of the next generation of bioprinted materials, playing an increasingly important role in cartilage regeneration, drug delivery, and personalized medicine. The continued development of multifunctional composites and intelligent biological scaffolds positions starch-based 3D-printed scaffolds as a promising frontier in tissue engineering and regenerative medicine.

## Figures and Tables

**Figure 1 molecules-30-03027-f001:**
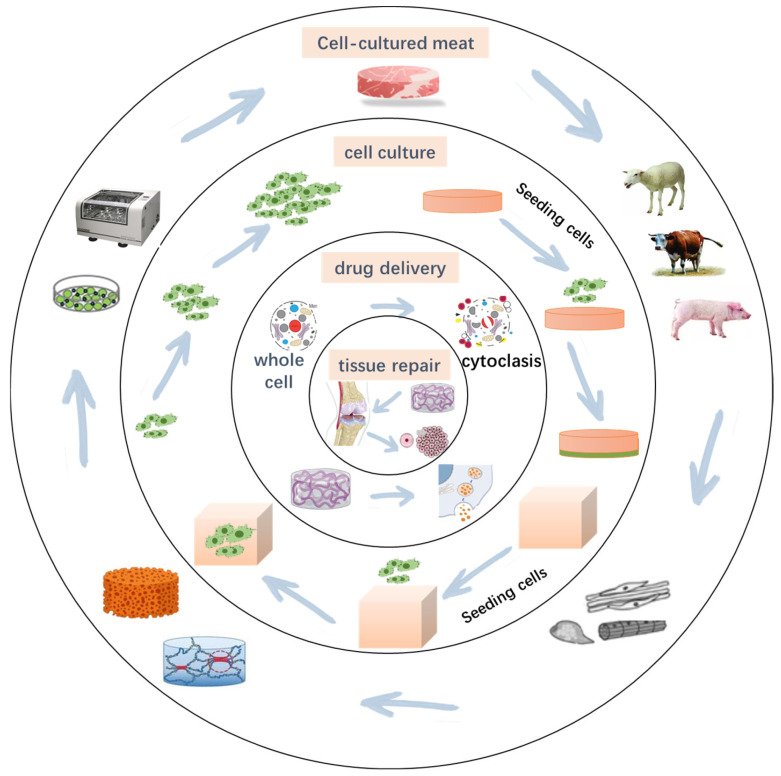
Application of 3D-printed scaffolds.

**Figure 2 molecules-30-03027-f002:**
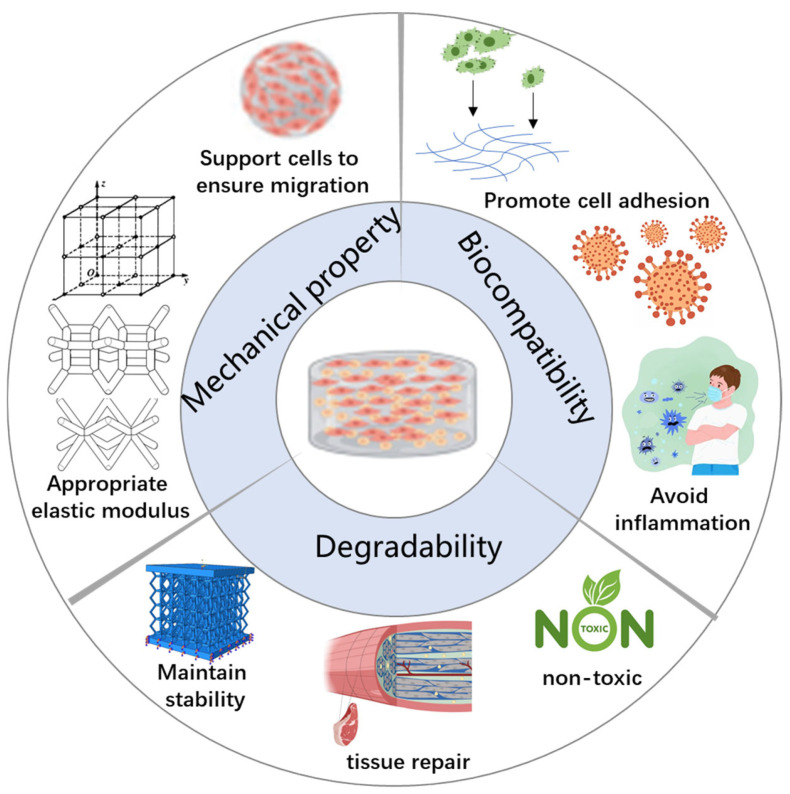
Characteristics of an ideal 3D bioprinted scaffold.

**Table 1 molecules-30-03027-t001:** The mechanical performance requirements for scaffolds imposed by different types of organizations.

Tissue Engineering	Mechanical Property	Typical Materials
Bone	It requires high compressive strength to support the proliferation of osteocytes and induce mineralization [13].	Calcium-phosphorus ceramics, starch-based composites, PLGA- nano-hydroxyapatite
Cartilage	High elasticity and moderate rigidity are required to adapt to the load of movement [30].	Gelatin/hyaluronic acid hydrogel, chitosan-based materials, elastin-based complexes
Muscle	Good elasticity and flexibility are required to support the stretching and contraction of cells [31].	Polyamino acid (PAA), silk fibroin
Vascular	Excellent elasticity and tensile strength are required to cope with blood flow pressure [32].	Polycaprolactone, elastin, polylactic acid-caprolactone copolymer (PLCL)

**Table 2 molecules-30-03027-t002:** Applications and Characteristics of Different Printing Technologies.

Printing Technology	Application Materials	Characteristic
Extrusion-based bioprinting (EBB)	Gelatin, alginate, chitosan, cellulose derivatives	Suitable for high-viscosity materials, it needs to solidify quickly to maintain the printed shape.
Inkjet bioprinting (IJP)	Hyaluronic acid, agarose, hydrogel	Suitable for low-viscosity materials, with high precision, it is mainly used for high-resolution patterned printing.
Stereolithography (SLA)	Methylacrylyl hyaluronic acid, PEG-DA	It requires photosensitive response materials, which can achieve ultra-high-precision printing, and is suitable for manufacturing fine structures.

**Table 3 molecules-30-03027-t003:** The properties and printing characteristics of different materials.

Materials	Biocompatibility	Mechanical Strength	Printing Adaptability	Degradability	Applications
Gelatin	√√√	✕	√√	√	Skin repair, cartilage engineering
Alginate	√√	✕	√√√	√	Soft tissue engineering, drug delivery
Hyaluronic acid	√√√	✕	√	√	Cartilage repair, cell culture

## Data Availability

No new data were created or analyzed in this study.

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
