# Peer review of "Applications and Recent Advances in 3D Bioprinting Sustainable Scaffolding Techniques"

_molecules, 2025, doi:10.3390/molecules30143027_

Round 1
Reviewer 1 Report
Comments and Suggestions for Authors
I would like to thank the authors for raising a very important topic, namely, biomaterials produced using 3D printing.
The authors focused on the characterization and application of various biomaterials: gelatin, sodium alginate, and starch. At the same time, they emphasized their contribution to the development of biomaterials.
I rate the entire article as very interesting. The authors cite 120 publications. The authors focused on selected materials and tried to refer to the latest publications available in the selected field.
However, in order for the article to be published, the following corrections are necessary:
Part 1:
- The abstract contains very general information about the article. Please add 2-3 sentences that will describe in detail and specify the new information contained in it.
- There are only 3 key terms added; please add more.
- Please increase the font in Figure 1, at the moment, the text is impossible to read.
- Line 34, please replace the word "quality" with a term that fits tissue engineering.
- Line 41, sentence: In cancer research and personalized medicine, bioprinted microenvironments simulate tumor niches, enabling drug screening and precision therapies." Please add a bibliography that supports this sentence.
- Line 48, please add a few sentences about your motivation for writing this review article. For example, "There is a lack of articles in the literature describing the latest technologies in application to...."
Paragraph 2, a very detailed description of the requirements for printed scaffolds.
- Line 172, please add a caption to the table.
- Line 172, please expand the table.
- Line 249, please add a caption to the table.
- I think it would be worth expanding the tables, which are a positive added value to the article. They make it easier for readers to understand complicated scientific terms.
- Please work on changing the conclusions, which should be concise and summarize the whole article. Please add information about trends in the future, how the authors think such studies can develop and what we can improve as researchers.
To sum up, the article is very interesting, but in order for it to be accepted, some corrections must be made.
Author Response
Comments 1: The abstract contains very general information about the article. Please add 2-3 sentences that will describe in detail and specify the new information contained in it.
Response 1: I agree with this comment. Therefore, I have revised and made the necessary additions. Mention exactly where in the revised manuscript this change can be found in line 32.
Comments 2: There are only 3 key terms added; please add more.
Response 2: I agree with this comment. Therefore, I have revised and made the necessary additions. Mention exactly where in the revised manuscript this change can be found in line 36. Keywords: starch; 3D bioprinting technology; scaffolds; natural macromolecular materials; cell culture.
Comments 3: Please increase the font in Figure 1, at the moment, the text is impossible to read.
Response 3: I agree with this comment. Therefore, I have revised. Mention exactly where in the revised manuscript this change can be found in line 50.
Comments 4: Line 34, please replace the word "quality" with a term that fits tissue engineering.
Response 4: I agree with this comment. Therefore, I have revised to characteristics. Mention exactly where in the revised manuscript this change can be found in line 58.
Comments 5: Line 41, sentence: In cancer research and personalized medicine, bioprinted microenvironments simulate tumor niches, enabling drug screening and precision therapies." Please add a bibliography that supports this sentence.
Response 5: I agree with this comment. Therefore, I have revised and made the necessary additions. Mention exactly where in the revised manuscript this change can be found in line 65.
Comments 6: Line 48, please add a few sentences about your motivation for writing this review article. For example, "There is a lack of articles in the literature describing the latest technologies in application to...."
Response 6: I agree with this comment. Therefore, I have revised and made the necessary additions. Mention exactly where in the revised manuscript this change can be found in line 79.
Comments 7 and 8: Line 172, please add a caption to the table. Line 172, please expand the table.
Response 7 and 8: I agree with this comment. Therefore, I have revised to “Table 1. The mechanical performance requirements for scaffolds imposed by different types of organizations”. Mention exactly where in the revised manuscript this change can be found in line 238.
Comments 9: Line 249, please add a caption to the table.
Response 9: I agree with this comment. Therefore, I have revised to “Table 2. Applications and Characteristics of Different Printing Technologies”. Mention exactly where in the revised manuscript this change can be found in line 335.
Comments 10: I think it would be worth expanding the tables, which are a positive added value to the article. They make it easier for readers to understand complicated scientific terms.
Response 10: I agree with this comment. However, the table in this article already covers the key materials and latest research that have been applied in the 3D printing field in recent years. If we were to expand further, we would mention the previously solved technologies, which might make the reading more difficult for the readers. If the reviewer still thinks that expansion would be better after considering my suggestions, I will proceed with the expansion.
Comments 11: Please work on changing the conclusions, which should be concise and summarize the whole article. Please add information about trends in the future, how the authors think such studies can develop and what we can improve as researchers.
Response 11: I agree with this comment. In Conclusion section, the first three paragraphs summarize the main materials of this article and conduct a comparative analysis, respectively elaborating on the advantages and disadvantages in different fields. After that, as “Future research directions for starch-based scaffolds include:” began to offer suggestions for future development trends. In order to facilitate readers' understanding, I have added some explanations in the text.

Reviewer 2 Report
Comments and Suggestions for Authors
The manuscript titled "Applications and Recent Advances in 3D Bioprinting Sustainable Scaffolding Techniques" is scientifically sound and well-structured, with extensive coverage of bioprinting materials. The manuscript would be suitable for publication in molecule if the comments below are answered and revised appropriately.
Q1. The abstract does not clearly articulate the novelty or scope of the review. The author should Include concrete findings (e.g., material performance rankings or specific recent innovations) would increase impact and readability.
Q2. The introduction provides a useful overview but lacks critical comparison of scaffold performance among material classes or recent meta-analytical insights from the literature. The author should consider citing quantitative comparisons or recent systematic reviews.
Q3. Statements about mechanical strength are qualitative (e.g., “high rigidity”). The author should include ranges of compressive modulus or elastic modulus values for scaffolds used in bone vs. cartilage engineering.
Q4. The section on 4D scaffolds is briefly inserted but lacks follow-through. Either expand this to link 4D behavior with tissue-specific applications (e.g., cartilage regeneration under mechanical load) or omit it for focus consistency.
Q5. The manuscript repeatedly highlights starch superiority, but some claims (e.g., “superior cell adhesion”) are not adequately supported by side-by-side comparisons. The author should include quantitative or comparative results from cited studies.
Q6. Figure 1 lacks scale and clarity. The circular layout illustrating scaffold applications is visually engaging but difficult to interpret. The author should add a clearer legend and possibly annotate arrows to indicate process flow or inter-relationships. Also. table on page 12 lacks reference citations. The table comparing gelatin, alginate, and hyaluronic acid lacks source references. It would be stronger if linked to specific studies or experimental benchmarks.
Q7. The conclusion heavily promotes starch-based scaffolds without sufficient balance. A more critical discussion of limitations (e.g., in vivo inflammatory response under different crosslinking methods) is needed for credibility. Also, the suggestions for future work are mostly material centric. The author should include translational gaps (e.g., regulatory barriers, GMP production, or sterilization challenges) to align better with real-world implementation.
Author Response
Comments 1: The abstract does not clearly articulate the novelty or scope of the review. The author should Include concrete findings (e.g., material performance rankings or specific recent innovations) would increase impact and readability.
Response 1: I agree with this comment. Therefore, I have revised and made the necessary additions. Mention exactly where in the revised manuscript this change can be found in line 32. “Through the comparison in this review, it can be observed that the starch scaffold not only has a lower price, but also can be modified to achieve multifunctionality, better meeting the performance requirements in more fields .”
Comments 2: The introduction provides a useful overview but lacks critical comparison of scaffold performance among material classes or recent meta-analytical insights from the literature. The author should consider citing quantitative comparisons or recent systematic reviews.
Response 2: I agree with this comment. Therefore, I have revised and made the necessary additions. Mention exactly where in the revised manuscript this change can be found in line 64. “The researchers found that the scaffolds prepared with chitosan could achieve a maximum tensile strength of 97 MPa in the dry state and 360% high strain at break in the wet state.”In line 68, “Liu used matrix/gelatin-sodium alginate scaffolds to prepare a biological scaffold with a cell survival rate of 90.79%±1.60%.”.
Comments 3: Statements about mechanical strength are qualitative (e.g., “high rigidity”). The author should include ranges of compressive modulus or elastic modulus values for scaffolds used in bone vs. cartilage engineering.
Response 3: I agree with this comment. Therefore, I have revised and made the necessary additions. Mention exactly where in the revised manuscript this change can be found in line 72. “He prepared a hydrogel with an elastic coefficient of 1.5 MPa using alginate and used it for drug delivery to treat breast cancer.”
Comments 4: The section on 4D scaffolds is briefly inserted but lacks follow-through. Either expand this to link 4D behavior with tissue-specific applications (e.g., cartilage regeneration under mechanical load) or omit it for focus consistency.
Response 4: I agree with this comment. Therefore, I have deleted.
Comments 5: The manuscript repeatedly highlights starch superiority, but some claims (e.g., “superior cell adhesion”) are not adequately supported by side-by-side comparisons. The author should include quantitative or comparative results from cited studies.
Response 5: I agree with this comment. Therefore, I have revised and made the necessary additions. In the comparison section, I have included a large number of references to support my conclusions.
Comments 6: Figure 1 lacks scale and clarity. The circular layout illustrating scaffold applications is visually engaging but difficult to interpret. The author should add a clearer legend and possibly annotate arrows to indicate process flow or inter-relationships. Also. table on page 12 lacks reference citations. The table comparing gelatin, alginate, and hyaluronic acid lacks source references. It would be stronger if linked to specific studies or experimental benchmarks.
Response 6: I agree with this comment. Therefore, I have revised and made the necessary additions. Figure 1 has been modified. Table 1 supplements the references can be found in line 240.
Comments 7: The conclusion heavily promotes starch-based scaffolds without sufficient balance. A more critical discussion of limitations (e.g., in vivo inflammatory response under different crosslinking methods) is needed for credibility. Also, the suggestions for future work are mostly material centric. The author should include translational gaps (e.g., regulatory barriers, GMP production, or sterilization challenges) to align better with real-world implementation.
Response 7: I agree with this comment. Therefore, I have revised and made the necessary additions. Mention exactly where in the revised manuscript this change can be found in line 839-848. “Because starch has a large number of active functional groups, the water resistance of starch scaffolds is insufficient. In many fields such as biological scaffolds, it is necessary to perform cross-linking modification on them to give them stronger water resistance. However, starch is widely available, for example, in corn, potatoes, wheat, etc., so the price is relatively low. In recent years, there have been more and more studies on starch modification, which also indicates that the large number of active hydroxyl groups carried by starch provides feasibility for its further research.”
